# Feasibility and Efficacy of Transcatheter Tricuspid Valve Repair in Patients with Cardiac Implanted Electrical Devices and Trans-Tricuspid Leads

**DOI:** 10.3390/jcm12154930

**Published:** 2023-07-27

**Authors:** Mhd Nawar Alachkar, Steffen Schnupp, Astrid Eichelsdoerfer, Andrea Milzi, Hesham Mady, Basem Salloum, Osama Bisht, Mohammed Cheikh-Ibrahim, Mathias Forkmann, Lukas Krygier, Christian Mahnkopf

**Affiliations:** 1Department of Cardiology and Angiology, Regiomed Klinikum Coburg, 3396450 Coburg, Germany; steffen.schnupp@regiomed-kliniken.de (S.S.); astrid.eichesldoerfer@regiomed-kliniken.de (A.E.); hesham.mady@regiomed-kliniken.de (H.M.); basem.salloum@regiomed-kliniken.de (B.S.); osama.bisht@regiomed-kliniken.de (O.B.); mohammed.cheikhibrahim@regiomed-kliniken.de (M.C.-I.); marhias.forkmann@regiomed-kliniken.de (M.F.); lukas.krygier@regiomed-kliniken.de (L.K.); christian.mahnkopf@regiomed-kliniken.de (C.M.); 2Department of Cardiology, Istituto Cardiocentro Ticino, 6900 Lugano, Switzerland; andrea.milzi@eoc.ch

**Keywords:** tricuspid regurgitation, transcatheter tricuspid valve repair, cardiac implanted electrical devices

## Abstract

Background: Transcatheter tricuspid valve repair using the edge-to-edge-technique (TEER) has emerged as an alternative therapy in patients with severe tricuspid regurgitation (TR) and high surgical risk. This study aimed to evaluate the feasibility and efficacy of tricuspid valve TEER in patients with cardiac implanted electric devices (CIEDs). Methods: All patients who underwent tricuspid valve TEER at our center were retrospectively included. Patients were classified according to the presence of CIEDs. Procedure success was defined as implantation of at least one clip and the reduction of TR of at least one grade. Procedure success and intrahospital outcome were compared between the two groups. Results: One-hundred and six consecutive patients underwent tricuspid TEER (age 80.1 ± 6.4 years, male = 42; 39.6%). Among them, 25 patients (23.6%, age 80.6 ± 7.3 years, male = 14; 56%) had CIEDs. Patients with CIEDs had a significantly lower left ventricular ejection fraction (LV-EF) compared to those without CIEDs (47.2 ± 15% vs. 56.2 ± 8.2%, *p* = 0.004, respectively). Moreover, arterial hypertension was more common in patients with CIEDs (96% vs. 79%, *p* = 0.048). The success of the procedure did not differ between the non-CIED vs. CIED group (93.8% vs. 92%, *p* = 0.748). Furthermore, the number and position of implanted clips, the duration of the procedure, the post-procedural pressure gradient across the tricuspid valve, and post-procedural TR severity were comparable between both groups. Conclusion: Tricuspid valve TEER is feasible and efficient in patients with CIEDs. The success of the procedure, as well as the intrahospital outcome were comparable between patients with and without CIEDs.

## 1. Introduction

Significant tricuspid valve regurgitation (TR) affects approximately 3% of the population aged more than 65 years [1]. Although being considered benign for a long time, recent data have shown that TR is associated with increased morbidity and mortality in patients with and without left ventricular dysfunction [2,3]. TR is particularly prevalent among patients with trans-tricuspid leads and cardiac electric implanted devices (CIEDs), and this may be due to different mechanisms, as it might be caused by the direct interaction between the lead and the leaflets, such as lead-induced impingement and restriction of the movement of the leaflets, or it might be secondary to the dilatation of the right heart chambers due to left heart disease [4,5]. The aging of the population has led to an increase in the prevalence of CIEDs [6]. Therefore, more patients with CIEDs and TR are being seen in clinical practice. Transcatheter treatment has emerged as an alternative to surgery in patients with relevant TR and high surgical risk. Among these, tricuspid valve repair, using the edge-to-edge technique (TEER), represents the most-common applied treatment option [7]. However, the presence of the trans-tricuspid lead may impede tricuspid valve TEER either due to direct interaction between the lead and valve leaflets or due to the echocardiographic shadow of the leads, complicating the visualization of the leaflets during the intervention [8]. Moreover, the presence of electrical leads may impede the manipulation of the device and the delivery system within the right atrium. This single-center retrospective study sought to evaluate the feasibility and efficacy of tricuspid TEER in patients with CIEDs.

## 2. Methods

Patients’ selection: All patients who underwent tricuspid valve TEER at our center were retrospectively included. The data were obtained from patients’ digital health records. All patients had symptomatic severe TR as defined by the guidelines of the European Society of Cardiology [9,10]. All patients were evaluated by the heart team, which included an interventional cardiologist, a cardiac surgeon, and an anesthetist, and were deemed to be non-operable. Patients with TR caused mainly by the CIED lead such as lead impinging one of the leaflets were deemed not appropriate for TEER. In all included cases, the multidisciplinary heart team recommended proceeding with tricuspid TEER. Patients were classified according to the presence of CIED into the CIED group and non-CIED group. The success of the procedure, post-procedural severity of TR, as well as intra-hospital outcome were compared between the two groups.

Transcatheter tricuspid valve repair: All procedures were performed using TriClip^®^ (Abbott) or Pascal^®^ (Edwards Life Science, Menlo Park, CA, USA) devices. All procedures were carried out under general anesthesia with the guidance of fluoroscopy and transesophageal echocardiography (Figure 1). The steps of performing tricuspid valve TEER are described in detail elsewhere [11]. In patients with CIEDs, the mid-esophageal bicaval view, as well as 3D-imaging in combination with fluoroscopy were used to deliver the clip without interfering with the lead. The transgastric en face short axis view was used to position the clip over the valve. The visualization of the leaflets during grasping was performed in an individual manner in every patient. More views were used to avoid the ultrasound shadowing caused by the lead and to ensure sufficient visualization of the leaflet such as the modified bicaval view, deep esophageal view, or grasping in transgastric en face short axis view. The strategy of performing TEER, including the positioning and rotation of the clip, was left to the operator’s preferences. Our approach will be described in the results.

Definition of outcome: The intrahospital outcome included the success of the procedure, defined as successful implantation of at least one clip and the reduction of TR of at least one grade, vascular complications defined as arteriovenous fistula, pseudo aneurysm, or relevant bleeding requiring operative or interventional management, and intrahospital mortality.

Statistical analysis: Continuous variables are expressed as the mean ± the standard deviation, and binary variables are expressed as a count (percentage). Patients were classified according to the presence of CIEDs into the non-CIED group and CIED group. Continuous variables were compared using independent sample *t*-test, and categorical variables were evaluated using Pearson’s chi-squared test. Statistical analyses were performed using SPSS Version 25.0 (IBM Corp., Armonk, NY, USA). Statistical significance was awarded by *p* < 0.05.

## 3. Results

Patients’ characteristics: The study included 106 consecutive patients. Among them, 25 patients (23.5%) had CIEDs. As mentioned before, in all patients with CIEDs, TR was secondary due to right atrial or ventricle dilatation and not directly caused by the lead of the device. Patients with TR caused mainly by the CIED lead such as the lead impinging one of the leaflets were deemed not appropriate for TEER. In the CIED group, 17 patients (68%) had a pacemaker, 5 patients (20%) had an implantable cardioverter-defibrillator (ICD), and 3 patients (12%) had cardiac resynchronization therapy (CRT). Among those patients, the lead position was assessed through the operating team and was found to be central in 14 patients and deep commissural (in the postero-septal commissure) in the other 11 patients. No lead was found to be positioned in the anteroseptal commissure. Patients in the CIED group had significantly more hypertension (96% vs. 79%, *p* = 0.048) and had a significantly lower left ventricular ejection fraction (47.2 ± 15% vs. 56.2 ± 8.2% *p* = 0.004). Otherwise, the patients’ characteristics were comparable in both groups (Table 1).

Intrahospital outcome: The procedure was successfully completed with the implantation of a minimum of one clip and the reduction of TR of at least one grade in 99 patients (93.4%). The Pascal^®^ (Edwards life science) device was used in 6 patients overall, including only 1 patient with CIED. The success of the procedure did not differ between groups (93.8% for the non-CIED group and 92% for the CIED group, *p* = 0.748). The duration of the procedure was also comparable between both groups. The number of implanted clips, the position of the clips, the pressure gradient across the tricuspid valve, as well as the post-procedural severity of TR did not differ between both groups. The occurrence of vascular complications also did not differ between the groups (8.6 in the non-CIED group vs. 12% in the CIED group, *p* = 0.616). Similarly, intrahospital mortality did not significantly differ between groups (9.9% vs. 4%, *p* = 0.357) (Table 2).

Strategy of TEER: In patients without CIEDs, the first clip was always positioned antero-septal. In patients with CIEDs, the positioning of the clip was decided according to the position of the lead, mostly starting antero-septal and eventually placing another clip either antero-septal, as in the case shown in Figure 1, or postero-septal (anteriorly or posteriorly to the lead) in patients with a central lead position. In patients with a commissural (postero-septal) lead position, the first clip was always positioned antero-septal, and eventually, another clip postero-septal (anteriorly to the lead) was implanted in patients with very severe TR or the clip was directly implanted postero-septal (anteriorly to the lead) in patients with relatively less-severe TR.

## 4. Discussion

The findings of our study advise that patients with CIEDs represent a considerable proportion of patients presenting with symptomatic TR and high surgical risk who are undergoing tricuspid TEER. In these patients, tricuspid valve TEER is feasible, and the results were comparable to those in patients without CIEDs.

TR in patients with CIEDs may be due to several mechanisms. In addition to direct interaction between the trans-tricuspid lead and leaflets of the tricuspid valve, left ventricular systolic dysfunction in patients with ICD or CRT may lead to right ventricle enlargement, causing secondary TR [12]. Furthermore, pacemaker-dependent patients may also develop systolic left ventricular dysfunction, eventually leading to right ventricular dilatation, dysfunction, and secondary TR [13]. Furthermore, asynchronized contraction of the right ventricle in pacemaker-dependent patients is also described as a potential etiology of TR in patients with pacemakers [3,4]. In our study population, patients with CIEDs had secondary TR and they had significantly lower ejection fractions than those without CIEDs.

In the Triluminate single-arm trial and in the Triluminate pivotal trial, patients with CIEDs represented 14% of the total cohort and 16% of the intervention group, respectively [14,15]. In the real-world TriValve registry, patients with CIEDs and trans-tricuspid leads represented 22.7% of all patients [7]. In our cohort, those patients represented 23.6% of all patients undergoing tricuspid TEER. A post hoc analysis of the above-mentioned trials and above-mentioned registry to investigate the long-term efficacy of tricuspid valve TEER in patients with CIEDs, in comparison to those without CIEDs, would be of interest.

In their single-center experience, Braun et al. reported similar outcomes in 18 patients with CIEDs in a cohort of 41 patients undergoing tricuspid TEER [16]. Similarly, Lurz et al. found no difference regarding the outcome of TEER in 33 patients with CIEDs out of 102 patients undergoing TEER. Furthermore, Lurz et al. reported no implication of tricuspid TEER on the function of CIEDs [17]. Our results confirmed those findings as the success rate and the reduction of TR were similar between patients with and without CIEDs in our study population.

Conclusively, our study found that tricuspid TEER is feasible in patients with trans-tricuspid leads, and it adds to our knowledge about this procedure in this group of patients.

Limitations: Although this study adds to the knowledge about tricuspid TEER in patients with CIEDs, we acknowledge that it is limited by its retrospective observational nature and lack of long-term follow up. In addition, this investigation suffered the usual shortcomings of a single-center study.

## 5. Conclusions

Tricuspid valve TEER is feasible and efficient in patients with CIEDs. The success of the procedure, as well as the intrahospital outcome were comparable between patients with and without CIEDs.

## Figures and Tables

**Figure 1 jcm-12-04930-f001:**
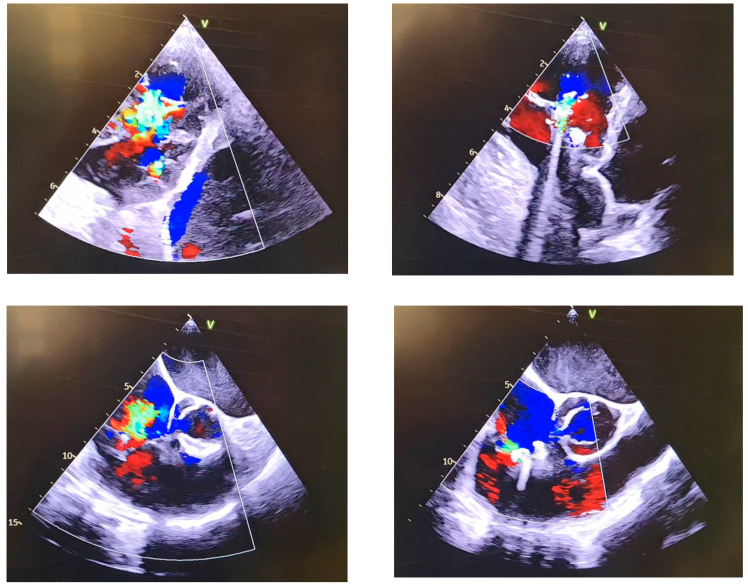
Transgastric en face view (**upper panel**) and mid-esophageal view (**lower panel**) of tricuspid valve in a patient with severe TR before and after TEER with implantation of two TriClip^®^ devices.

**Table 1 jcm-12-04930-t001:** Clinical characteristics of the study population.

	All	Non-CIED (n = 81)	CIED (n = 25)	*p*-Value
*Clinical characteristics*				
Age, years	80.1 ± 6.4	79.9 ± 6.1	80.6 ± 7.3	0.321
Male, n (%)	42 (39.6)	28 (34.6)	14 (56)	0.055
DM, n (%)	29 (27.3)	19 (23.5)	10 (40)	0.105
HTN, n (%)	88 (83)	64 (79)	24 (96)	0.048
Atrial fibrillation, n (%)	94 (88.6)	73 (90.1)	21 (84)	0.398
Chronic renal failure, n (%)	83 (78.3)	60 (74.1)	23 (92)	0.057
Dialysis, n (%)	4 (0.4)	4 (4.9)	0 (0)	0.257
PAD	9 (8.4)	8 (9.9)	1 (4)	0.357
COPD	17 (16)	14 (17.3)	3 (12)	0.529
CAD, n (%)	60 (42.1	42 (51.9)	18 (72)	0.076
Previous PCI, n (%)	92 (86.7)	69 (85.2)	23 (92)	0.379
Previous CABG, n (%)	11 (10.3)	7 (8.7)	4 (16)	0.292
Previous other cardiac surgery, n (%)	10 (9.4)	8 (9.9)	2 (8)	0.779
Previous mitral valve TEER, n (%)	26 (24.5)	17 (20.9)	9 (36)	0.127
Previous aortic valve intervention	3 (2.8)	3 (3.8)	0 (0)	0.329
NYHA class				0.931
I	1	1	0	
II	23	18	5	
III	65	49	16	
IV	6	5	1	
Type of CIED				
Pacemaker, n (%)	-	-	17 (68)	-
ICD, n (%)	-	-	5 (20)	-
CRT, n (%)	-	-	3 (12)	-
TR severity				0.498
I	-	-	-	
II	-	-	-	
III	60 (56.6)	46 (56.8)	14 (56)	
IV	42 (39.6)	31 (38.3)	11 (44)	
V	4 (3.8)	4 (4.9)	0 (0)	
LV-EF,%	54 ± 10.9	56.2 ± 8.2	47.2 ± 15	0.004
TAPSE, mm	20.7 ± 5.1	20.5 ± 5	21.4 ± 5.7	0.223

CIED: cardiac implanted electrical device, DM: diabetes mellitus, HTN: arterial hypertension, PAD: peripheral artery disease, COPD: chronic obstructive pulmonary disease, CAD: coronary artery disease, PCI: percutaneous coronary intervention, TEER: transcatheter edge-to-edge repair, LVEF: left ventricular ejection fraction, NYHA: New York Heart Association, TR: tricuspid regurgitation.

**Table 2 jcm-12-04930-t002:** Intrahospital outcome of tricuspid valve TEER in patients with and without cardiac implanted electrical devices.

	All	Non-CIED (n = 81)	CIED (n = 25)	*p*-Value
Success of the procedure	99 (93.4)	76 (93.8)	23 (92)	0.748
Number of clips	1.42 ± 0.6	1.41 ± 0.6	1.44 ± 0.5	0.803
Position of clip				0.09
Antero-septal	75 (70.7)	57 (70.4)	18 (72)	
Postero-septal	17 (16)	12 (14.8)	5 (20)	
both	14 (13.2)	12 (14.8)	2 (8)	
Pmean, mmHG	2.8 ± 1.2	2.8 ±1.2	2.9 ± 1.2	0.811
Time of procedure, min	90.2 ± 36	92.1 ± 37.9	83.4 ± 30.1	0.324
TR severity				0.961
I	47 (44.3)	37 (45.7)	10 (40)	
II	50 (47.2)	37 (45.7)	13 (52)	
III	4 (3.8)	3 (3.7)	1 (4)	
IV	4 (3.8)	3 (3.7)	1 (4)	
V	1 (0.9)	1 (1.2)	0 (0)	
Vascular complications, n (%)	10 (9.4)	7 (8.6)	3 (12)	0.616
Intrahospital mortality, n (%)	9 (8.4)	8 (9.9)	1 (4)	0.357

CIED: cardiac implanted electrical devices, TEER: transcatheter edge-to-edge repair, Pmean: pressure mean gradient across tricuspid valve, TR: tricuspid regurgitation.

## Data Availability

Data are available in REGIOMED Klinikum Coburg.

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
