# Peer review of "Feasibility and Efficacy of Transcatheter Tricuspid Valve Repair in Patients with Cardiac Implanted Electrical Devices and Trans-Tricuspid Leads"

_jcm, 2023, doi:10.3390/jcm12154930_

Round 1
Reviewer 1 Report
The original manuscript by Alachkar and colleagues presents data on the outcome of patients with TEER of TR depending on the presence of intracardiac transtricuspid leads.
The manuscript is well written and provides interesting data, especially as there is still a huge respect of many cardiologists for tv TEER in presence of transtricuspid leads.
Clinical experience shows us, that positioning of the leads leading to a restriction of the anterior or posterior tv leaflet is associated with worse procedural outcome and clip positioning is more difficult or even impossible. Did you also include or at least stratify your patients concerning the lead placement and did these patients have a worse TEER outcome? Please present these results and discuss. If possible, could you also provide statistical analysis.
Could you also explain your interventional approach?
Did you always place the first clip in the anteroseptal commissure working your way to the centre of the valve.
Did you find a difference between the two products for TEER used in the study? Please provide information on that.
After adressing these smaller issues, the manuscript has merit to my mind and addresses an important problem in current cardiology providing a relatively large sample size of patients with transtricuspid leads
The original manuscript by Alachkar and colleagues presents data on the outcome of patients with TEER of TR depending on the presence of intracardiac transtricuspid leads.
The manuscript is well written and provides interesting data, especially as there is still a huge respect of many cardiologists for tv TEER in presence of transtricuspid leads.
Clinical experience shows us, that positioning of the leads leading to a restriction of the anterior or posterior tv leaflet is associated with worse procedural outcome and clip positioning is more difficult or even impossible. Did you also include or at least stratify your patients concerning the lead placement and did these patients have a worse TEER outcome? Please present these results and discuss. If possible, could you also provide statistical analysis.
Could you also explain your interventional approach?
Did you always place the first clip in the anteroseptal commissure working your way to the centre of the valve.
Did you find a difference between the two products for TEER used in the study? Please provide information on that.
After adressing these smaller issues, the manuscript has merit to my mind and addresses an important problem in current cardiology providing a relatively large sample size of patients with transtricuspid leads and TEER.
Author Response
- The original manuscript by Alachkar and colleagues presents data on the outcome of patients with TEER of TR depending on the presence of intracardiac transtricuspid leads.
The manuscript is well written and provides interesting data, especially as there is still a huge respect of many cardiologists for tv TEER in presence of transtricuspid leads.
Authors' reply: We are grateful for the reviewer for reviewing this manuscript and for his positive feedback. We will reply to the comments of the reviewer in a point-to-point manner and we would address the changes in the revised manuscript by writing them in red.
Clinical experience shows us, that positioning of the leads leading to a restriction of the anterior or posterior tv leaflet is associated with worse procedural outcome and clip positioning is more difficult or even impossible. Did you also include or at least stratify your patients concerning the lead placement and did these patients have a worse TEER outcome? Please present these results and discuss. If possible, could you also provide statistical analysis.
Author's reply: We thank the reviewer for calling our attention to this point. As we mentioned in our manuscript, in all patients with CIED, TR was secondary and not directly caused by the lead due to impinging of the leaflets as an example. We emphasized this information in our revised manuscript (page 2, lines 61-62 and page 3, lines 95-98). Furhtermore, we added the informations about position of the lead to our revised manuscript (page 3, lines 100-103).
However, we believe, that dividing the small sample of patients with CIED further into smaller subgroups with analyzing and comparing the outcome between those subgroups would be underpowered.
Could you also explain your interventional approach?
Did you always place the first clip in the anteroseptal commissure working your way to the centre of the valve.
Authors' reply: we thank the reviewer for his valuable comment. As suggested we described our approach in performing TEER in both groups in our revised manuscript (page 5, lines 125-133).
- Did you find a difference between the two products for TEER used in the study? Please provide information on that.
Authors' reply: we thank the reviewer for his valuable comment. The Pascal® (Edwards life science) device was used in overall 6 patients, including only one patient with CIED, whereas TriClip® (Abbott) was used in all other patients. Due to the very small sample size of patients treated with Pascal®,compared to those treated with TriClip®, no statistical analysis was performed to compare the two products. In our revised manuscript we supplied the above mentioned informations (Page 4, lines 115-116).
As suggested by the reviewer, more details were added to the introduction as well as to the methods.
Reviewer 2 Report
In this paper Alachkar et al. retrospectively analysed their results of TEER procedures in patients with and without CIEDs.
It is well known how TEER has emerged as a valid alternative therapy in patients with TR who are at high surgical risk. In patients with previous CIED implantation, this procedure could be more challenging because of technical aspects leading to non-optimal results.
The paper is well structured and the introduction gives a clear overview about the state of the art. Patient selection and characteristics are well explained.
Methods: it could be useful to explain procedural solutions that allow the operator to successfully manage TEER implantation despite CIED lead-related ultrasound shadows and to avoid lead during implantation.
Statistical analysis has no evident issue.
Results are clear, adding interesting evidence to the contemporary literature.
Conclusions are fully supported by the results.
Language is clear and English is fine, without stylistic and/or syntactical flaws.
Overall, this manuscript from Alachkar et al. is a well done paper that gives further evidence about a relatively new procedure, with the vision of expanding the cohort of patients eligible for this procedure.
Author Response
In this paper Alachkar et al. retrospectively analysed their results of TEER procedures in patients with and without CIEDs.
It is well known how TEER has emerged as a valid alternative therapy in patients with TR who are at high surgical risk. In patients with previous CIED implantation, this procedure could be more challenging because of technical aspects leading to non-optimal results.
The paper is well structured and the introduction gives a clear overview about the state of the art. Patient selection and characteristics are well explained.
Methods: it could be useful to explain procedural solutions that allow the operator to successfully manage TEER implantation despite CIED lead-related ultrasound shadows and to avoid lead during implantation.
Statistical analysis has no evident issue.
Results are clear, adding interesting evidence to the contemporary literature.
Conclusions are fully supported by the results.
Language is clear and English is fine, without stylistic and/or syntactical flaws.
Overall, this manuscript from Alachkar et al. is a well done paper that gives further evidence about a relatively new procedure, with the vision of expanding the cohort of patients eligible for this procedure.
Authors' reply: We are very thankful for the reviewer for reviewing the manuscript and for the positive feedback. We added further description of performing the procedure in to our methods (page 2, lines 70-76)